# Antimicrobial, Antiviral, and In-Vitro Cytotoxicity and Mosquitocidal Activities of *Portulaca oleracea*-Based Green Synthesis of Selenium Nanoparticles

**DOI:** 10.3390/jfb13030157

**Published:** 2022-09-19

**Authors:** Amr Fouda, Waad A. Al-Otaibi, Taisir Saber, Sahar M. AlMotwaa, Khalid S. Alshallash, Mohamed Elhady, Naglaa Fathi Badr, Mohamed Ali Abdel-Rahman

**Affiliations:** 1Department of Botany and Microbiology, Faculty of Science, Al-Azhar University, Nasr City, Cairo 11884, Egypt; 2Department of Chemistry, College of Science and Humanities, Shaqra University, Shaqra 11961, Saudi Arabia; 3Department of Clinical Laboratory Sciences, College of Applied Medical Sciences, Taif University, P.O. Box 11099, Taif 21944, Saudi Arabia; 4College of Science and Humanities-Huraymila, Imam Mohammed Bin Saud Islamic University (IMSIU), Riyadh 11432, Saudi Arabia; 5Department of Zoology and Entomology, Faculty of Science (Girls’ Brunch), Al-Azhar University, Nasr City, Cairo 11751, Egypt

**Keywords:** plant-based Se-NPs, *Portulaca oleracea*, *Candida* spp., HAV, Cox-B4, human hepatocellular carcinoma, *Culex pipiens*

## Abstract

The aqueous extract of *Portulaca oleracea* was used as a biocatalyst for the reduction of Na_2_SeO_3_ to form Se-NPs that appeared red in color and showed maximum surface plasmon resonance at a wavelength of 266 nm, indicating the successful Phyto-fabrication of Se-NPs. A FT-IR chart clarified the role of plant metabolites such as proteins, carbohydrates, and amino acids in capping and stabilizing Se-NPs. TEM, SAED, and XRD analyses indicated the formation of spherical, well-arranged, and crystalline Se-NPs with sizes in the range of 2–22 nm. SEM-EDX mapping showed the maximum peaks of Se at 1.4, 11.3, and 12.4 KeV, with weight and atomic percentages of 36.49 and 30.39%, respectively. A zeta potential of −43.8 mV also indicated the high stability of the synthesized Se-NPs. The Phyto-synthesized Se-NPs showed varied biological activities in a dose-dependent manner, including promising activity against pathogenic bacteria and *Candida* species with varied MIC values in the range of 12.5−50 µg·mL^−1^. Moreover, the Se-NPs showed antiviral activity toward HAV and Cox-B4, with percentages of 70.26 and 62.58%, respectively. Interestingly, Se-NPs showed a target orientation to cancer cell lines (HepG2) with low IC_50_ concentration at 70.79 ± 2.2 µg·mL^−1^ compared to normal cell lines (WI−38) with IC_50_ at165.5 ± 5.4 µg·mL^−1^. Moreover, the as-formed Se-NPs showed high activity against various instar larvae I, II, III, and IV of *Culex pipiens*, with the highest mortality percentages of 89 ± 3.1, 73 ± 1.2, 68 ± 1.4, and 59 ± 1.0%, respectively, at 50 mg L^−1^. Thus, *P. oleracea*-based Se-NPs would be strong potential antimicrobial, anti-viral, anti-cancer, and anti-insect agents in the pharmaceutical and biomedical industries.

## 1. Introduction

Diseases caused by pathogenic bacteria, fungi, viruses, and insects have become major problems for governments, the public, and regulatory agencies, because they are considered the main causes of death worldwide [1,2]. The construction of new compounds with multifunctional antimicrobial, antiviral, anticancer, antioxidant, and anti-insect properties is challenging. Fortunately, nanotechnology has provided a wide range of nanoparticle compounds that are compatible with various biomedical and biotechnological applications [3]. These materials can be used in pharmaceuticals, cosmetics, smart textiles, wastewater treatment, antimicrobial agents, antiviral agents, anti-insect agents, antioxidants, drug delivery, drug release, the food industry, paints, paper preservation, heavy metal sorption, dye removal, etc. [4,5,6,7,8]. Although nanomaterials such as silver, gold, copper, magnesium zinc, cadmium, and gadolinium have varied biomedical applications, they are limited due to either high production costs (e.g., silver, gold, copper, and magnesium) or their toxic nature (e.g., titanium, gadolinium, and cadmium) [9].

Recently, selenium nanoparticles (Se-NPs) have received more attention towards incorporation into the biomedical sector due to their high biocompatibility and low toxicity [1]. Selenium is known as an essential element for normal functions of the immune system in humans and animals and preventing degenerative and lethal diseases [10]. Further, it is important as an enzyme co-factor, and selenoproteins protect the cells from oxidative stresses [11]. Researchers have reported that the nutrient-containing selenium can be used to ameliorate hepatic diseases caused by alcohol consumption, the accumulation of toxic substances such as heavy metals, and the effects of chemotherapeutic drugs [11,12]. For optimal human health, adults require nearly 40−300 mg/day of a nutritional supplement containing selenium [13]. However, the addition of selenium material as it stands in nutritional supplements has several disadvantages, including low adsorption and low bioavailability; therefore, the use of Se-NPs in these additives can overcome these problems [14]. In addition, several investigations have reported that Se-NPs have anti-viral activities against various types of viruses, such as the influenza virus and the Hepatitis B virus, through the inhibition of the replication and transcription of DNA [14]. Further, Se-NPs have shown promising activity against a wide range of pathogenic microbes such as *Staphylococcus aureus*, *Bacillus cereus*, *Salmonella typhi*, *Listeria monocytogenes*, *Klebsiella pneumonia*, *Escherichia coli*, *Burkholderia cenocepacia*, *Pseudomonas aeruginosa*, *Stenotrophomonas maltophilia*, *Achromobacter xylosoxidans*, *Candida albicans*, *Candida tropicalis*, *Candida glabrata*, *Aspergillus flavus*, *Aspergillus niger*, and *Aspergillus fumigates* [4,15]. Along with the above, Se-NPs have shown anti-cancer activities against a wide range of cancer cell lines and various types of insect vectors [5].

Different approaches are used to fabricate Se-NPs, including chemical, physical, and biological approaches. In most cases, chemical and physical methods are disfavored because of their high cost, biosafety concerns, and toxic byproducts and the harsh conditions needed for synthesis. Therefore, biological synthesis has been recommended as an alternative method for coping with these disadvantages. Biogenic synthesis incorporates natural products secreted by different biological entities (bacteria, fungi, yeast, actinomycetes, plants) as a reduction and then coating the surface of nanomaterials to increase stability and reduce the aggregation or agglomeration of NPs with time [16,17,18]. Plant-based green synthesis of Se-NPs is preferred over other methods because it is inexpensive and eco-friendly, and it increases the stability of NPs by increasing their coating agents [19]. Various plant extracts have been used for the fabrication of various nanoparticles. For instance, Ag-NPs, Au-NPs, Pd-NPs, Pt-NPs, CuO-NPs, ZnO-NPs, TiO_2_-NPs, etc. are fabricated using plant aqueous extract [20,21,22]. Few investigations have reported the efficacy of plant extract in the biogenic synthesis of Se-NPs.

In the current study, an aqueous extract of *Portulaca oleracea* L. was used to fabricate Se-NPs for the first time. UV-Vis spectroscopy, X-ray diffraction (XRD), Fourier transform infrared spectroscopy (FT-IR), Transmission Electron Microscopy (TEM), and Scanning Electron Microscopy combined with an Energy-Dispersive X-ray (SEM-EDX) were used to characterize the Phyto-synthesized Se-NPs. Their antimicrobial activities against different pathogenic prokaryotic and eukaryotic organisms (Gram-positive bacteria, Gram-negative bacteria, and *Candida* spp.) were evaluated. Further, their antiviral activity against the hepatitis A virus (HAV) and Coxsackie B virus (Cox-B4); in vitro cytotoxicity against human hepatocellular carcinoma (HepG2) and human normal lung fibroblast (WI-38) cell lines; and anti-insect activity were also investigated.

## 2. Materials and Methods

### 2.1. Materials

The metal precursor (sodium selenite, Na_2_SeO_3_) used for the green synthesis of Se-NPs was analytical grade (Sigma Aldrich, Darmstadt, Germany). Antibacterial activity was achieved using coded bacterial strains purchased from the American Type Culture Collection (ATCC), and anti-*Candida* activity was assessed using various clinical strains that were obtained from the Microbiology Laboratory, National Research Centre, Dokki, Giza, Egypt. In vitro cytotoxicity was achieved using HepG2 (cancer cell) and WI-38 (normal cell) lines obtained from ATCC, and anti-viral activity was assessed toward HAV and Cox-B4 that were purchased from the Holding Company for Biological Products and Vaccines (VACSERA), Dokki, Giza, Egypt.

### 2.2. Plant-Mediated Green Synthesis of Se-NPs

#### 2.2.1. Preparation of Leaf Aqueous Extract of *P. oleracea* L.

The leaves of *P. oleracea* were collected, washed three times with distilled H_2_O, left to dry at room temperature, and finally ground to form a powder. Approximately 5 g of the leaf powder was mixed with 100 mL of distilled H_2_O and heated at 50 °C under stirring conditions for 60 min, followed by centrifugation at 5000 rpm for 15 min to collect a supernatant that was used as a biocatalyst for the fabrication of Se-NPs.

#### 2.2.2. *P. oleracea* Aqueous Extract Mediated Phyto-Fabrication of Se-NPs

Three millimolar of Na_2_SeO_3_ were prepared in 90 mL distilled H_2_O before being completed to 100 mL with the collected plant extract. The pH of this mixture was adjusted at 8 by 1 N NaOH before being subjected to stirring for one hour at 38 °C. Finally, the mixture was left overnight in a dark environment. The formation of Se-NPs was checked by the color change from pale green to red [5]. The leaf aqueous extract in the absence of the metal precursor was used as control. The final nanoparticle product was collected, washed three times with distilled H_2_O, and oven-dried at 300 °C for 2 h.

### 2.3. Characterization of Phyto-Fabricated Se-NPs

#### 2.3.1. UV-Vis Spectroscopy

The reduction of Na_2_SeO_3_ by metabolites in the leaf aqueous extract of *P. oleracea* was first checked by its color change. The absorbance of the synthesized red color was monitored by measuring the wavelength in the range of 200–800 nm to detect the surface plasmon resonance that is specific for Se-NPs. Absorbance was measured using a JENEWAY spectrophotometer (JENWAY-6305, Staffordshire, UK).

#### 2.3.2. X-ray Diffraction (XRD)

The phase structure of the Phyto-synthesized Se-NPs was detected by XRD using a 0.5 mm glass capillary tube in the two-theta range of 10–70°. The analysis was achieved by X’ Pert Pro (Philips, Eindhoven, the Netherlands) connected with Cu, which serves as an X-ray source. The radiation (Cu Kα) was produced at λ = 1.54 Å by adjusting the operating system at 40 kV and 30 mA.

The Debye–Scherrer equation was used to calculate the average crystal size as follows [1]:(1)La=Kλ/βcosθ
where La is the average crystal size, K is the constant and equal to 0.9, λ is the wavelength of the radiation source (λ = 1.54 Å), and β is the full width at a half maximum in 2θ value.

#### 2.3.3. Fourier Transform Infrared (FT-IR) Analysis

The role of various functional groups in the *P. oleracea* leaf aqueous extract in the bioreduction and stabilizing of Se-NPs was investigated by FT-IR (Cary 630 FTIR model, Tokyo, Japan). In this analysis, 250 mg of Phyto-synthesized Se-NPs was mixed with KBr and pressed to form a disk that was subjected to scanning at a wavenumber of 400–4000 cm^−1^ [23].

#### 2.3.4. Transmission Electron Microscopy (TEM)

The morphological characteristics such as size, shape, and agglomeration of Phyto-synthesized Se-NPs were investigated using TEM (TEM JEOL 1010, Tokyo, Japan). Briefly, 1 g of synthesized Se-NPs was dissolved in 2 mL ethanol and sonicated for 10 min. After that, a few drops of suspension were loaded on a Cu-grid and stand-up to complete adsorption. The loaded Cu-grid was dabbed on blotting paper to remove excess solution before being subjected to image capture [24].

#### 2.3.5. Scanning Electron Microscopic-Energy-Dispersive X-ray (SEM-EDX)

The surface morphology and elementary mapping of the synthesized Se-NPs were assessed by SEM-EDX analysis (JEOL, JSM-6360LA, Tokyo, Japan). The synthesized Se-NPs were loaded on holders followed by coating with gold by a sputter coater in a vacuum. The SEM image was captured at 30 kV. The SEM was connected with an EDX instrument that was used for the qualitative and quantitative elemental composition of the sample [1].

#### 2.3.6. Dynamic Light Scattering (DLS) and Zeta Potential

The size distribution and hydrodynamic size of the Phyto-synthesized Se-NPs in the colloidil solution were investigated by DLS. The as-formed sample was resuspended in high-purity solvent (MiliQ H_2_O) to avoid a shadow on the signal during particle scattering [25]. The charge of the Se-NPs’ surface was detected using the Malvern Zetasizer instrument (Nano-ZS, Malvern, UK).

### 2.4. Antimicrobial Activity

The efficacy of Phyto-synthesized Se-NPs in suppressing the growth of human pathogenic Gram-positive bacteria, Gram-negative bacteria, and the clinical *Candida* species was investigated using the agar well diffusion method [26]. Tested pathogenic bacteria were represented by *Staphylococcus aureus* ATCC-6538, *Bacillus subtilis* ATCC-6633, *Escherichia coli* ATCC-8739, and *Pseudomonas aeruginosa* ATCC-9022, and the clinical *Candida* isolates were identified as *C. albicans*, *C. glabrata*, *C. tropicalis*, and *C. parapsilosis* in the National Research Centre, Cairo, Egypt. At first, the bacterial and *Candida* strains were refreshed on appropriate broth media (nutrient broth for bacteria and Sabouraud dextrose broth for *Candida*) and incubated for 24 h at 35 ± 2 °C. After that, approximately 50 µL of each strain (OD = 1.0) were inoculated onto 100 mL sterilized Muller–Hinton agar media, well shaken, and poured under aseptic conditions into Petri dishes. After solidification, three wells (each with a diameter of 0.7 mm) were prepared in the inoculated plates before being filled with 100 µL of the Se-NP concentration (300 µg/1 mL DMSO) and set in the refrigerator for 1 h before incubating at 35 ± 2 °C for 24 h. After incubation, the appearance of the clear zone (mm) around each well was recorded [27]. In the same manner, the inhibitory effect of various concentrations (200, 100, 50, 25, 12.5, and 5.25 µg·mL^−1^) of Se-NPs was assessed to identify the value of the minimum inhibitory concentration (MIC) for each strain. The experiment was performed in triplicate.

### 2.5. Antiviral Activity

#### 2.5.1. Determination of Cytotoxicity of Se-NPs on Vero Cells

To evaluate the maximum non-toxic concentration, Vero cells, as normal cells, were implemented in a 96-well, flat-bottomed microtiter plate at a density of 10^4^ cells/well containing a 100 µL growth medium and incubated overnight at 37 °C in a 5% CO_2_ incubator for cell attachment. When the confluent sheet of the Vero cells was formed, the growth medium was discarded, and the wells were washed twice with washing media. Subsequently, the attached cells were treated with different concentrations of Se-NPs, ranging from 1000 µg·mL^−1^ to 31.25 µg·mL^−1^ per well. Double fold dilution of Se-NPs was performed in DMSO, and 100 µL of each dilution was tested in triplicate, whereas the cells only received a complete medium as a control. The plates were incubated in a humidified incubator at 37 °C in 5% CO_2_ and visualized frequently for up to 48 h, and any physical signs of toxicity, such as partial or complete monolayer destruction, roundness, cellular granulation, and membrane shrinkage, were recorded. After that, 20 µL of prepared MTT solution (5 mg mL^−1^ in PBS) was added to each well and mixed with a shaker at 150 rpm for 5 min. The plates were then incubated at 37 °C, in a 5% CO_2_ incubator for 4 h. Following incubation, the growth medium was discarded from the wells and the formazan crystals were resuspended in 200 µL of DMSO and mixed thoroughly by shaking at 150 rpm for 5 min. The optical density of each well was determined at 560 nm, and the background was subtracted at 620 nm. The maximum non-toxic concentration (MNTC) of the examined Se-NPs was determined using the following formula:(2)Cell toxicity(%)=cell viability(%)−100

The percentages of cell toxicity were plotted on the *Y*-axis, and the concentration of the sample was plotted on the *X*-axis, which allowed for obtaining the MNTC as the concentration of Se-NPs needed to keep the viability of the host cell with no significant difference to the untreated control, and it was utilized for further antiviral activity. The cytotoxic concentration needed to reduce cellular growth by a 50% (CC_50_) value of the sample was calculated by regression analysis.

#### 2.5.2. Antiviral Assay of Se-NPs

To assess the effect of Se-NPs on the inhibition of the hepatitis A virus (HAV) and Coxsackie B virus (Cox-B4) infectivity, Vero cells were implemented in 96-well, flat-bottomed microtiter plates at a density of 10^4^ cells/well containing 200 µL growth medium and allowed to adhere overnight at 37 °C in 5% CO_2_. A virus suspension was incubated with non-lethal concentrations of Se-NPs (1:1, *v*/*v*) at room temperature for 1 h. After incubation, 100 µL from the viral/sample suspension was added to the well implemented with Vero cells, whereas three wells were considered non-infected cells (control) and contained Vero cells and growth media only. The plates were mixed on a shaker for 5 min at 150 rpm followed by incubation for 24 h at 37 °C in 5% CO_2_ to allow the virus to take effect. The cellular viability of infected and non-infected Vero cells was conducted using the absorbance values of formazan crystals used in the MTT reagent as described for the cytotoxicity assay. The anti-HAV and anti-COX-B4 activities were determined by measuring the difference in the values between the optical density of infected and uninfected cellular viabilities.

### 2.6. In-Vitro Cytotoxicity of Se-NPs on Cancer (HepG2) and Normal (WI-38) Cell Lines

In vitro assessment of Se-NP’s cytotoxicity against human hepatocellular carcinoma (HepG2) and human normal lung fibroblast (WI-38) cell lines was performed using a 3-(4,5-dimethylthiazol-2-yl)-2,5-diphenyl tetrazolium bromide (MTT) assay. The cells were obtained from the American Type Culture Collection. Next, 1 × 10^5^ cells/mL were seeded in 96 well plates and incubated at 37 °C in 5% CO_2_ for 24 h. Then, the culture medium was changed, and six different consecrations of the Se-NPs (1000, 500, 250, 125, 62.5, and 32.25 µg·mL^−1^) were added and then incubated for 24 h. After that, the culture medium was removed and washed with phosphate buffer saline (PBS). The cellular viability of treated and untreated cells was conducted using the absorbance values of formazan crystals used in the MTT reagent, as described for the cytotoxicity assay. Finally, the absorbance was measured at 560 nm using a microplate reader. The experiments were conducted in triplicate. Cell viability percentages were calculated according to the following equation [28]:(3)$$\mathrm{Cell}\;\mathrm{viability}(\%)=\frac{\mathrm{Absorbance}\;\mathrm{of}\;\mathrm{experimental}\;\mathrm{sample}}{\mathrm{Absorbance}\;\mathrm{of}\;\mathrm{experimental}\;\mathrm{control}}\times 100$$

### 2.7. Mosquitocidal Activity

#### 2.7.1. Larvae Rearing

Different instar larvae (I, II, III, and IV) of *Culex pipiens* were collected from Medical Entomology, Giza, Egypt. The collected larvae were added to a plastic cup filled with tap water and reared in the Entomology Laboratory, Faculty of Science, Al-Azhar University according to the standard protocol [29]. The experiment was conducted at 27 ± 2 °C, a relative humidity (RH) of 70–80%, and a photoperiod of 12: 12 light/dark. Larvae feeding was achieved with dog biscuits mixed with yeast and added to the larvae under laboratory conditions.

#### 2.7.2. Bioassay

The potentiality of Phyto-synthesized Se-NPs to control various instar larvae of *C. pipiens* was achieved according to the WHO standard method [30]. In this bioassay, 25 *C. pipiens* larvae from each instar were moved to a cup filled with 100 mL of different concentrations of Se-NPs (10, 20, 30, 40, and 50 mg L^−1^). A cup filled with 100 mL of dechlorinated tap water was used as a control. The experiment was conducted in five replicates for each Se-NP concentration. The larvae that lost their ability to reach the surface of the treatment solution after cup disturbances were calculated as dead. The mortality percentages were calculated after 48 h using the following equation [31].
(4)$$\mathrm{Mortality}\;\mathrm{percentages}(\%)=\frac{\mathrm{mortality}\;\mathrm{in}\;\mathrm{treatment}-\mathrm{mortality}\;\mathrm{in}\;\mathrm{control}}{100-\mathrm{mortality}\;\mathrm{in}\;\mathrm{control}}\times 100\%$$

### 2.8. Statistical Analysis

The obtained data were analyzed by the statistical package SPSS v17 and represented as the means of three independent replicates. The difference between treatments was measured by *t*-test or ANOVA test followed by the Tukey HSD test at *p* < 0.05. The mortality percentages of different instar larvae were measured by probit analysis, with LC_50_ and LC_90_ calculated using Finney’s method.

## 3. Result and Discussion

### 3.1. Plant-Based Biogenic Synthesis of Se-NPs

The biogenic fabrication method for NP production by the green approach is preferred for its simplicity, biocompatibility, ease of precipitate, eco-friendliness, cost-effectiveness, and avoidance of toxic byproducts during chemical synthesis and harsh conditions required for the physical method [32]. Moreover, biogenic synthesis using plant extracts has various advantages over other biological entities, such as microorganisms, due to its biosafety as compared to some microorganisms, rapid single step, huge metabolites secreted by plants that increase reduction and NP stability, and cost-effectiveness [33]. In the current study, the leaf aqueous extract of *P. oleracea* was used to fabricate Se-NPs. Sodium selenite (metal precursor) was colorless and then converted to a red color after mixing with plant extract as a result of the reduction of Na_2_SeO_3_ to elemental selenium (Se^0^). Gunti et al. reported that the formation of a red color is related to the excitation of SPR that indicates the reduction of Na_2_SeO_3_ to Se^0^ [1]. The red color was formed once plant extract was added to the metal precursor solution and the pH value was adjusted. Interestingly, the color intensity increased with time because of the complete reduction; therefore, the color absorbance in the current study was measured after 24 h. Srivastava and Mukhopadhyay reported that the change in color intensity with time affects the intensity of SPR but does not affect their location [34]. Further, the color intensity of fabricated Se-NPs by the aqueous extract of *Brassica oleracea* increased with time and reached the maximum after 48 h, indicating the complete reduction of SeO_3_^2−^ to Se^0^ [35]. The reduction of Na_2_SeO_3_ to Se-NPs can be attributed to the presence of various metabolites such as carbohydrates, phenolic compounds, alkaloids, tannins, flavonoids, phenolic compounds, and triterpenoids [36].

The main factor affecting the bio-reduction of metal precursors is pH value. In the current study, the optimum pH value for the fabrication of Se-NPs was 8.0 (this value was selected based on the optimization of pH values on the color intensity). The obtained data indicated that the functional groups that existed in the plant aqueous extract were more active in alkaline conditions. In this condition, the capping agents coating the NPs’ surfaces were more stable, leading to the enhancement of the NPs’ stability and minimizing aggregations, as reported previously [37,38]. To the best of our knowledge, this is the first report on the biogenic synthesis of Se-NPs using the leaf aqueous extract of *P. oleracea*.

### 3.2. Characterization of Se-NPs

#### 3.2.1. UV-Vis Spectroscopy Analysis

The bio-reduction of Se-NPs was checked by a color change, i.e., the formation of red color was monitored by UV-Vis spectroscopy at a wavelength of 200–800 nm. Data showed that the maximum absorption peak noticed at 266 nm was responsible for the SPR of Se-NPs (Figure 1). The data obtained show the efficacy of plant extracts used as a biocatalyst for the reduction of SeO_3_^2^^−^ to Se^0^. The absorption peak in the current study was to be compatible with those reported for the fabrication of Se-NPs by plant extract. For instance, the maximum SPR peak for Se-NPs synthesized by the leaf aqueous extract of *Diospyros montana* was observed at 261 nm [39]. Further, the maximum absorption peak of Se-NPs fabricated by fruit aqueous extract of *Emblica officinalis* was shown at 270 nm [1]. Fesharaki et al. reported that the maximum absorption peak related to SPR for green-synthesized Se-NPs was in the range of 200–300 nm [40].

#### 3.2.2. X-ray Diffraction (XRD)

The phase structure (crystalline or amorphous nature) was detected by XRD analysis. As shown in Figure 2A, the Phyto-synthesized Se-NPs had nine X-ray plans (100), (101), (110), (102), (111), (201), (003), (202), and (210) that corresponded to the Bragg reflection at two-theta values of 23.6°, 29.7°, 43.6°, 45.3°, 51.6°, 55.7°, 61.5°, 71.6, and 78.4°, respectively. The diffraction peaks obtained refer to the formation of the crystalline structure of Se-NPs and are matched with the standard card of JCPDS No. 06-0362 [5,38]. These data are compatible with those obtained for the crystalline Se-NPs obtained using leaf aqueous extract of *Withania somnifera* [41]. The unassigned diffraction peaks in the XRD pattern could be attributed to the various biomolecules that are present in aqueous plant extract [42,43]. The average crystallite size was calculated according to the Debye–Scherrer equation. In the current study, the average crystallite size of Se-NPs was 32 nm. Similarly, the average crystallite size of Se-NPs formed using a flower aqueous extract of *Cassica auriculata* was calculated using the Debye–Scherrer equation and recorded at 30.7 nm [44].

#### 3.2.3. Fourier Transform Infrared (FT-IR) Analysis

The various functional groups present in plant aqueous extract and their activity in the bio-reduction, capping, and stabilizing of Se-NPs were investigated using FT-IR. As shown, The FT-IR chart of aqueous extract contains five adsorption peaks at a wavenumber of 3410, 2080, 1632, 1184, and 530 cm^−1^ (Figure 2B). The broadness and strong peak at 3410 cm^−1^ signify N–H and O–H groups of amino acids in proteins. This peak was shifted to 3440 cm^−1^ in the Se-NPs chart [45]. The peak at a wavenumber of 2080 cm^−1^ could be attributed to the carbohydrate moiety in the plant extract. A peak at 1632 cm^−1^ corresponding to the C=O group overlapped with the stretching N–H vibration group of the polysaccharide that exists in plant extract [1,46], which shifted to 1640 cm^−1^ after the fabrication of Se-NPs. The other peaks in the chart of the plant extract, 1184 cm^−1^ and 530 cm^−1^, corresponded to the stretching C–N of amine and C–l of the halo compound, respectively. Additional peaks at 860, 1100, 1370, 2140, and 2930 cm^−1^ were observed in the FT-IR chart due to the interaction of capping agents in *P. oleracea* extract with Se-NPs (Figure 2B). The medium peak at 2930 cm^−1^ signifies the stretching C–H of alkane, whereas the strong peak at 2140 cm^−1^ refers to the stretching S–C≡N of thiocyanate [47]. The peaks at wavenumbers 1370, 1100, and 860 cm^−1^ corresponded to a NO_3_ stretching deformation and the asymmetric stretching groups of C–O–C and C–N, respectively [48,49]. The stretching C–O–S that overlapped with NO_3_ ions was distinguished at a wavenumber of 860 cm^−1^ [47,50]. According to the FT-IR analysis, it can be concluded that the plant metabolites, such as carbohydrates, proteins, and amino acids, coated the surface of synthesized Se-NPs and protected them from aggregations, leading to an increase in the Se-NPs’ stability.

#### 3.2.4. Transmission Electron Microscopy (TEM)

The main factors affecting the biomedical and biotechnological activities of NP are size, shape, surface area, dispersion, coating agents, elementary mapping, and stability [51,52]. TEM and SEM-EDX analyses are useful techniques in investigating these characteristics. In the current study, the leaf aqueous extract of *P. oleracea* has the efficacy to fabricate the spherical shape of Se-NPs without any aggregation (Figure 3A). The TEM image showed that the size of the Phyto-synthesized Se-NPs was in the ranges of 2–22 nm, with an average diameter of 10.6 ± 4.2 nm (Figure 3B). In support of our investigation, fruit aqueous extract of *Vitis vinifera* was used to fabricate spherical Se-NPs with a size range of 3–18 nm [53]. Further, the alcoholic leaf extract of *Psidium guajava* exhibited the potential to fabricate Se-NPs with spherical shapes and sizes in a range of 8–20 nm [41]. The TEM-selected area electron diffraction (TEM-SAED) for *P. oleracea*-based Se-NPs was displayed as a bright spot corresponding to the hexagonal crystallographic structure of as-formed Se-NPs [54]. Various published studies report that the biological activities of Se-NPs are shape- and size-dependent, and the smaller sizes have higher activity. For instance, the antioxidant activity of three different shapes (cubes, spheres, and rods) of biogenic-synthesized Se-NPs was greater for cubic shapes with percentages of 65.0 ± 2.5%, followed by spheres and rods, with percentages of 52.5 ± 2.5 and 47.5 ± 2.5%, respectively. The spheric shapes showed promising antibacterial activity, followed by cubic and rod shapes [55]. On the other hand, the antimicrobial activity of synthesized Se-NPs from garlic aqueous extract was greater for those with sizes of 21–40 nm than for those with larger sizes of 41–50 nm [56]. Therefore, in the current study, the high activity of Phyto-synthesized Se-NPs can be predicted due to their smaller sizes.

#### 3.2.5. Scanning Electron Microscopic-Energy-Dispersive X-ray (SEM-EDX)

The elemental composition of as-formed Se-NP was detected by SEM-EDX (Figure 3D,E). As shown, the Phyto-synthesized Se-NPs were well-dispersed without agglomeration. The Se ions are represented by three absorption peaks at different bending energies of 1.4, 11.3, and 12.4 KeV, which refer to the peak of SeLα, SeKα, and SeKβ, respectively [57]. In agreement with our study, the absorption peaks of Se-NPs fabricated by the extract of *Vitis vinifera* were observed at binding energies of 1.4 KeV (SeLα-peak), 11.2 KeV (SeKα-peak), and 12.5 KeV (SeKβ-peak) [53]. Analysis of the EDX chart showed that the Se ions represented the maximum weight percentages in the sample, with a value of 36.49%. The maximum absorption peak at a bending energy of 0.2 KeV indicates the C ions that have weight percentages of 30.68%. Moreover, a peak at 0.5 KeV is characteristic of O ions. Other additional peaks for Cl and K are present with low weights and atomic percentages with values of (4.27 and 11.07%) and (2.02 and 4.75%), respectively. The presence of these additional peaks can be attributed to the scattering of capping and stabilizing agents that coated the surface of Se-NPs during EDX analysis [58]. Sharma et al. reported that the presence of oxygen and carbon peaks in the synthesized sample indicates the presence of alkyl chains in the stabilizing agents [53].

#### 3.2.6. Dynamic Light Scattering (DLS) and Zeta Potential

The hydrodynamic size of plant-based Se-NPs as well as the size distribution in colloidal solution was investigated by DLS. As shown, the average size of Se-NPs fabricated by aqueous extract of *P. oleracea* was 68 nm, according to the graph of the size distribution (Figure 4A). Similarly, the average size of Se-NPs fabricated by bacterial strain *Ralstonia eutropha* calculated by DLS was 70.9 nm [34]. Further, Se-NPs synthesized by aqueous extract of *Ceropegia bulbosa* have an average size of 55.9 nm based on DLS analysis [5]. Accordingly, it can be concluded that the sizes calculated by DLS were larger than those obtained by other techniques such as TEM and XRD. This finding could be attributed to the fact that the sizes calculated by DLS were highly affected by coating agents that interfered with the measurement. In addition, the DLS measures the hydrodynamic residues of synthesized particles. The DLS calculation size can be affected by the non-homogenous distribution of particles in colloidal solution [59,60].

The stability of Phyto-synthesized Se-NPs was investigated by zeta potential analysis. Figure 4B shows that the negative zeta potential value at −43.8 mV indicates the high stability of synthesized Se-NPs. In a similar study, the zeta potential value of Se-NPs synthesized by a coded Gram-negative strain of *E. coli* ATCC-35218 was −42.5 mV, indicating high stability, as reported by the authors [61]. The stability degree according to zeta potential values was detected as follows: ±0–10 mV is very unstable; values in the ranges of ±10–20 mV are not very stable; ±20–30 mV is stable; and values greater than ± 30 mV are highly stable [62]. The zeta potential result for Se-NPs synthesized in the current study indicates that the surface charge was negative. This negative value could be related to the presence of the reducing agent’s polyphenolic and flavonoid compounds in the plant extract, which displays electrostatic forces in green-synthesized NPs [63]. If all particles in the colloidal solution have a negative or positive zeta potential value, then the particles tend to repel each other, thereby avoiding aggregation. On the other hand, if some particles bear a positive zeta potential and others bear a negative value, then the particles tend to aggregate together [57]. As shown in Figure 4B, all particles in the synthesized solution have a negative zeta potential value, and this provided high stability without aggregation.

### 3.3. Antimicrobial Activity

Bacterial and *Candida* diseases are considered major causes of mortality and morbidity worldwide. Further, the appearance of antibiotic-resistant microbes causes major problems in medicine. Therefore, it is urgent to discover new compounds characterized by safe-to-use, cheap, and broad-spectrum activity. The amazing advances in nanotechnology science in the last decade can offset this gap due to the unique antimicrobial properties of nanoparticles, especially those synthesized by green approaches. In the current study, the inhibitory action of Phyto-synthesized Se-NPs was investigated against various human pathogenic isolates using the agar well diffusion method. The data obtained proved the hypothesis that the antimicrobial activity of NPs was dose-dependent, and the zone of inhibition was decreased by lowering the dose concentration (Figure 5 and Figure 6). The synthesized Se-NPs showed broad spectrum activity against Gram-positive bacteria, Gram-negative bacteria, and unicellular fungi at high concentrations. As shown, the maximum inhibition zones were recorded at 300 µg·mL^−1^ with values of 16.0 ± 1.0, 16.7 ± 0.6, 17.7 ± 0.7, 16.7 ± 0.6, 15.6 ± 0.6, 19.3 ± 0.8, 18.3 ± 0.6, and 16.7 ± 0.7 mm for *Bacillus subtilis*, *Staphylococcus aureus*, *Pseudomonas aeruginosa*, *Escherichia coli*, *Candida albicans*, *Candida glabrata*, *Candida tropicalis*, and *Candida parapsilosis*, respectively (Figure 5 and Figure 6). These inhibition zones were decreased to 13.7 ± 0.6, 14.3 ± 0.6, 14.3 ± 0.6, 14.4 ± 0.6, 13.7 ± 0.6, 17.3 ± 0.7, 17.3 ± 0.6, and 15.3 ± 0.6 mm, respectively, at lower concentrations. Similar to our study, Se-NPs fabricated by the bacterial strain *B. subtilis* showed promising antimicrobial activity against Gram-positive bacteria, Gram-negative bacteria, and different species of *Candida* and *Aspergillus* [4]. The authors reported that the diameter of inhibition zones gradually increased with concentration and recorded the highest diameter at 500 µg·mL^−1^. Zonaro et al. reported that there is a correlation between the antimicrobial activity of Se-NPs and their size [64]. The activity increased with smaller sizes, and this is attributed to the increase in the surface-to-volume ratio, with small size leading to enhanced NP biological reactivity.

The minimum inhibitory concentration (MIC) that is required to inhibit the growth of various microorganisms should be detected for the integration of the Phyto-synthesized Se-NPs in medical sectors. In the current study, the MIC value of as-formed Se-NPs varied according to organisms. It was 50 and 25 µg·mL^−1^ for *B. subtilis* and *S. aureus* with a zone of inhibition of 9.7 ± 0.6 and 8.7 ± 0.6 mm, respectively (Figure 5). The MIC value for Gram-negative bacteria was 12.5 µg·mL^−1^, with clear zones of 8.3 ± 0.6 and 8.2 ± 0.3 mm for *P. aeruginosa* and *E. coli*, respectively (Figure 5). Accordingly, the Gram-positive bacteria were more resistant to Se-NPs in lower concentrations than Gram-negative bacteria. Compatible with the current study, *Emblica officinalis*-based Se-NPs exhibited MIC values against Gram-positive and Gram-negative bacteria in the range of 9.2 ± 0.8–59.8 ± 2.6 µg·mL^−1^ [1]. The MIC value of the synthesized Se-NPs toward various *Candida* spp. was 25 µg·mL^−1^ for *C. albicans* and *C. parapsilosis*, whereas it was 12.5 µg·mL^−1^ for *C. glabrata* and *C. tropicalis* (Figure 6). Overall, *Candida* spp. are more sensitive toward Phyto-synthesized Se-NPs than prokaryotic species, as reported previously [38]. Moreover, *P. aeruginosa* was more sensitive, followed by *E. coli* and Gram-positive bacteria. This phenomenon can be attributed to the thin peptidoglycan layer of Gram-negative bacteria [41,65]. The inhibitory mechanism of plant-based Se-NPs might be related to the generation of reactive oxygen species (ROS) that disrupt the integrity of the plasma membrane and hence suppress the selective permeability function. Moreover, the inhibitory action of biogenic-synthesized Se-NPs can be attributed to the destruction of the cell wall, followed by binding with the cell membrane and entering the cytoplasm that leads to a change or mutation in metabolism cycles, DNA replications, protein synthesis, interaction with the thiol group or sulfhydryl group in amino acids, and protein structures that lead to denaturation and cell death [14,64,66]. The anti-*Candida* activity of Se-NPs could be attributed to their efficacy in destroying the sterol profile in the cell wall of *Candida* by disrupting the biosynthesis of the ergosterol pathway [67,68]. Interestingly, ergosterol is an important component of the homogeneity, rigidity, fluidity, and integrity of the plasma membrane [69].

### 3.4. Antiviral Activity

Before the assessment of the antiviral activity of the Phyto-synthesized Se-NPs, the cytotoxicity of the sample on the host cell (Vero normal cells) was studied and illustrated in Figure 7. The antiviral effect of the sample should be active against the viruses without inducing remarkable toxicity on the host cells. The visible cytotoxicity of the Se-NPs was observed on the treated Vero cells in the range of 1000−125 µg·mL^−1^, and a half cytotoxic concentration CC_50_ of Se-NPs was observed at 224.58 µg·mL^−1^. The toxicity was evidenced in the form of inhibition of the cellular growth rate, rounding, clumping, and detachment of the cells from the surface. As anticipated, when the concentration of Se-NPs decreased to 62.5 µg·mL^−1^, no visible toxicity on Vero cells was observed; therefore, this was used as the maximum non-toxic concentration of Se-NPs to evaluate their antiviral activity. The results illustrated that HAV- and Cox-B4-infected Vero cells without MNTC of Se-NPs revealed 1.79- and 2.05-fold inhibition in their viabilities, respectively, when compared with those Vero cells infected with HAV- and Cox-B4 suspensions pre-incubated with non-lethal concentrations of the sample. The Se-NPs could protect the HAV- and Cox-B4-infected cells from the virus, with a percent protection of 70.25% and 62.5%, respectively, and could increase the cell viability to 84.2% and 76.5%, respectively (Table 1). These observations suggested that the Se-NPs resisted the proliferation of the viruses in Vero cells and inhibited the activation of apoptotic proteins by viruses inside the host cells [70]. In addition, Se was reported as a modulator for specific enzymes responsible for some key biological interactions and ROS elimination [71]. It was previously reported that the deficiency in selenium levels could increase susceptibility to viral infection [72].

### 3.5. Effect of Se-NPs on Cytotoxicity and Cellular Morphology of Normal WI-38 and Cancer HepG2 Cells

The anticancer efficacy of biosynthesized Se-NPs was evaluated against human normal lung fibroblast (WI-38) and human hepatocellular carcinoma (HepG2). Both normal and cancer cells were treated with different concentrations (32.25–1000 µg·mL^−1^) of Se-NPs for 24 h, and the cytotoxicity was determined by MTT assay. Figure 8 exhibits that Se-NPs induced cytotoxicity in a dose-dependent manner in WI-38 and HepG2 cells. At a lower concentration of the biosynthesized Se-NPs (31.25−62.5 µg·mL^−1^), there was almost no cytotoxic effect on normal cells, whereas a noticeable reduction in cell viability in cancer cells was observed. Moreover, HepG2 cells exerted more than a two-fold rise in the cytotoxicity of Se-NPs in comparison to WI-38 cells. The IC_50_ values of Se-NPs were 70.79 ± 2.22 µg·mL^−1^ for HepG2 and 165.53 ± 5.37 µg·mL^−1^ for WI-38 cells. The results obtained indicated that the Se-NPs were more effective in killing cancer cells, with an elicited minimal cytotoxicity effect on normal cells. Therefore, the higher anticancer effect of Se-NPs at a lower dose implies that biosynthesized Se-NPs may promote a synergistic effect with chemotherapeutic drugs for improving cancer treatment. In a recent study, Se-NPs synthesized by the fungal strain *Penicillium corylophilum* exhibited an IC_50_ value against cancer cell lines of Caco-2 with a value of 171.8 µg·mL^−1^, as compared to a low IC_50_ value (104.3 µg·mL^−1^) against normal cell lines WI-38 [73]. Moreover, the concentration of Phyto-synthesized Se-NPs using an aqueous extract of *Ceropegia bulbosa* that leads to a 50% death in normal breast cells (HBL-100) was twice the concentration that induces apoptosis in breast cancer (MDA-MB-231) [5]. The minimal effect of lower concentrations of Se-NPs on normal cells might be attributed to the normal redox balance [10].

The morphological changes in WI-38 and HepG2 cells after being treated with Se-NPs for 24 h at different concentrations are depicted in Figure 9. In comparison to the untreated control, the HepG2 and WI-38 cells revealed morphological alterations including cells becoming more spherical, a decrease in cell count and confluency, and a loss of contact with a neighboring cell. In agreement with MTT findings, the HepG2 at a lower concertation (62.5 µg·mL^−1^) revealed cellular damage and increased the space between neighboring cells in comparison with untreated cells. On the contrary, the same concentration of WI-38 cells showed no morphological alteration. In agreement with the results obtained, spherical Se-NPs with a size of 60 nm fabricated by laminarin polysaccharides showed promising activity in inducing toxicity in HepG2 cells [74]. The authors reported that the Se-NPs have the efficacy to activate various mitochondrial pathways that enhance apoptosis and suppress the autophagy mechanism. Further, Se-NPs decrease or stop the expression of Bcl-2 (anti-apoptotic factor) and reduce the negative effects of Bcl-2 on Beclin-1 in HepG2 cell lines. In a recent study, Se-NPs fabricated by leaf aqueous extract of *Asteriscus graveolens* inhibited the proliferation of HepG2 cells through the damage of DNA, leading to arrest of the cell cycle [75].

It was previously proved that selenium can promote cancer cell apoptosis with minimal adverse effects on normal cells [10,76]. The probable cause behind the anticancer activity of Se-NPs is that the selenium NPs can enhance inflammation and oxidative stress by generating ROS, leading to decreased cell viability percentages [10]. The main route for the entrance of Se-NPs into the cells is through endocytosis. The malignant cells are characterized by redox imbalance. The presence of Se-NPs inside the cell triggers the generation of free radicals, which causes various destructive mechanisms. Among these mechanisms are the disruption of the mitochondrial membrane and endoplasmic reticulum stress. The disruption to the mitochondrial membrane causes protein leakage and hence enhances apoptosis via caspase activation. On the other hand, the stresses due to free radicals would activate different molecular pathways such as the MAPK/Erk, Wnt/β-catenin, PI3K/Akt/mTOR, VEGF, and NF-κB. The latter pathway (NF-κB) is responsible for the disruption of cellular homeostasis, and the other pathways are essential in oncogenic signaling. The disruption of these pathways due to the action of Se-NPs leads to the impairment of cellular proliferation and the suppression of the signaling responsible for growth promotion in the tumor microenvironment. All these destructive mechanisms lead to the end of DNA fragmentation, causing arrest of the cell cycle and cell death [10,77].

### 3.6. Mosquitocidal Activity of Se-NPs

*P. oleracea*-based Se-NPs showed promising larvicidal activity through high mortality percentages of various instar larvae of *Culex pipiens* at different concentrations (Table 2). Data analysis implies a positive relationship between the dose of Se-NPs and the mortality rate: the mortality increased as the concentration increased for different instar larvae. At a high concentration of Se-NPs (50 mg L^−1^), the mortality percentages were 89.0 ± 3.1, 73 ± 1.2, 68 ± 1.4, and 59 ± 1.0% for I, II, III, and IV instar larvae, respectively. These mortality percentages were decreased by decreasing the dose to reach 55.0 ± 2.0, 35.0 ± 1.1, 31.3 ± 1.5, and 27.3 ± 1.5% at a lower concentration of 10 mg L^−1^ for the same sequence of instar larvae (Table 2). In agreement with the current study, the larvicidal activity of *Ceropegia bulbosa*-based Se-NPs against various instar larvae of *Aedes albopictus* was dose-dependent [5]. The authors reported that the mortality percentages were 82.0 ± 3.1, 64.0 ± 1.2, 59.0 ± 1.4, and 48.0 ± 1.0% for the first, second, third, and fourth instar larvae of *Aedes albopictus*, respectively, at a concentration of 50 mg L^−1^, and these percentages decreased to 57.0 0 ± 2.0, 39.0 ± 1.1, 32.3 ± 1.5, and 28.3 ± 1.5%, respectively at 10 mg L^−1^. In the same regard, the activity of Se-NPs fabricated by *Murraya koenigii* extract against fourth instar larvae of *Aedes aegypti* were dose-dependent and caused mortality percentages of 99.2, 77.6, 58.3, and 42.8% for a concentration of 10, 7.5, 5, and 2.5 µg·mL^−1^, respectively [78]. Analysis of variance revealed that the LC_50_ values of plant-mediated, green-synthesized Se-NPs against the first to fourth *C. pipiens* instar larvae were 1.12, 18.4, 25.4, and 28.8 mg L^−1^, respectively, whereas the LC_90_ values were 42.4, 59.3, 66.4, and 69.4 mg L^−1^, respectively (Table 2). Data recorded by Sowndarya et al. are incompatible with the current study; the former reported that the LC_50_ and LC_90_ of *Clausena dentata*-based Se-NPs against *Anopheles stephensi*, *Aedes aegypti*, and *Culex quinquefasciatus* were (240.7, 104.1, and 99.6 mg L^−1^) and (367.01, 158.1, and 153.6 mg L^−1^), respectively [79]. On the contrary, the LC_50_ and LC_90_ values of plant-mediated biosynthesis of Se-NPs against the fourth instar larvae of *A. aegypti* were 3.5 and 8.1 µg·mL^−1^, respectively [78]. The varied values of mortality percentages (LC_50_, and LC_90_) of plant-based Se-NPs could be attributed to the difference in secondary metabolites secreted by plants that act as capping and stabilizing agents [78,80]. In another study, the authors reported that the phytochemical groups, such as terpenoids, steroids, phenolic compounds, essential oils, and alkaloids, secreted by different plants have a major role in insecticidal activities [81]. This concept confirmed our hypothesis that the activity of plant-based Se-NPs has a correlation with plant secondary metabolites.

The mosquitocidal activity of plant-mediated, green-synthesized Se-NPs could be attributed to their toxic effects on various cellular pathways. The toxicity of Se-NPs against various vectors is due to their efficacy to denature main cell components as well as essential enzymes, which leads to a disruption of membrane permeability and hence blocks the synthesis of energy in the cell (ATP), ultimately leading to cell death [82,83]. Another toxicity mechanism of plant-based Se-NPs against different vectors is their potential to bind with sulfur-containing amino acids and phosphorus-containing compounds such as nucleic acids, leading to the destruction of these main cell components [17].

## 4. Conclusions

In the current study, Se-NPs were fabricated by harnessing the active metabolites in the leaf aqueous extract of *P. oleracea*. The Phyto-synthesized Se-NPs were characterized by a color change from no color to red, UV-Vis spectroscopy that showed maximum SPR at 266 nm, FT-IR that exhibited the role of plant metabolites in Phyto-fabrication, and TEM, XRD, and EDX analyses that revealed the successful formation of spherical, crystallographic particles without aggregation and with an average size of 10.6 ± 4.2 nm. The plant-based Se-NPs showed antimicrobial activities against pathogenic Gram-positive bacteria, Gram-negative bacteria, and various clinical unicellular fungi, with varied inhibition zones based on concentration. Further, the Se-NPs showed promising antiviral activity toward HAV and Cox-B4, with percentages of 70.26 and 62.58%, respectively. Moreover, they showed anti-cancer activity against human hepatocellular cancer (HepG2) at a low IC_50_ value of 70.79 ± 2.2 µg·mL^−1^ as compared to the IC_50_ value of normal cell lines (165.5 ± 5.4 µg·mL^−1^). Interestingly, the Phyto-synthesized Se-NPs exhibited promising anti-insect properties against various instar larvae of *C. pipiens*, with mortality percentages of 89 ± 3.1, 73 ± 1.2, 68 ± 1.4, and 59 ± 1.0% for I, II, III, and IV instar larvae, respectively, at 50 mg L^−1^. Our data reveal the biopotential impacts of plant extract to fabricate spherical Se-NPs with promising multi-biological activities to be incorporated into biomedical applications.

## Figures and Tables

**Figure 1 jfb-13-00157-f001:**
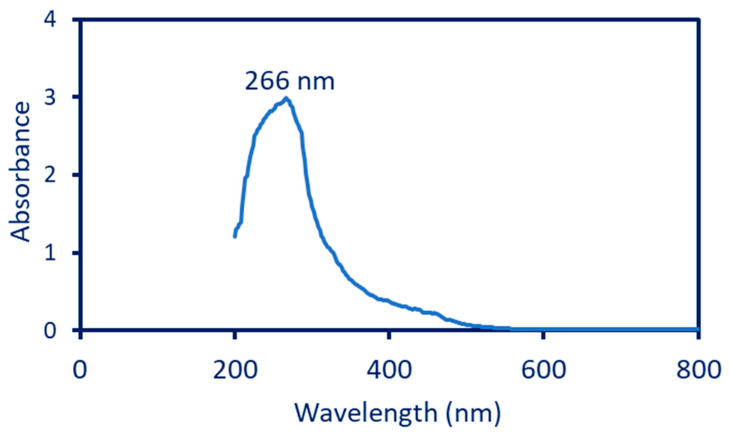
UV-Vis spectroscopy of the Phyto-synthesized Se-NPs shows the maximum absorption peak at 266 nm.

**Figure 2 jfb-13-00157-f002:**
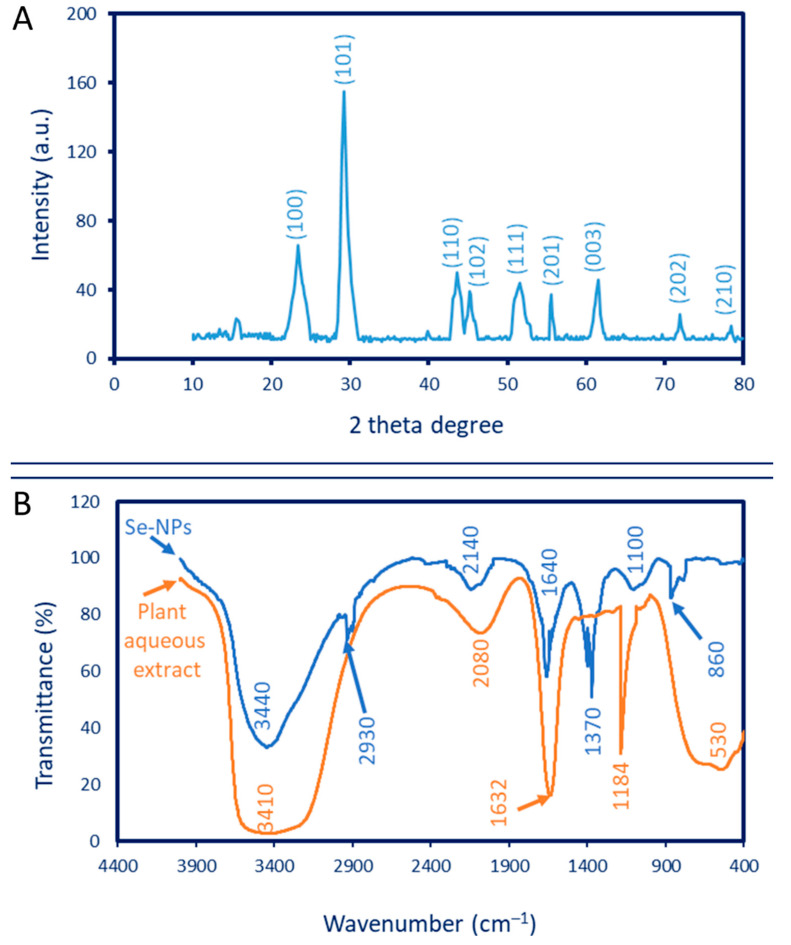
Characterization of *P. oleracea*-based, green-synthesized Se-NPs. (**A**) XRD analysis shows the crystalline nature, (**B**) FT-IR chart.

**Figure 3 jfb-13-00157-f003:**
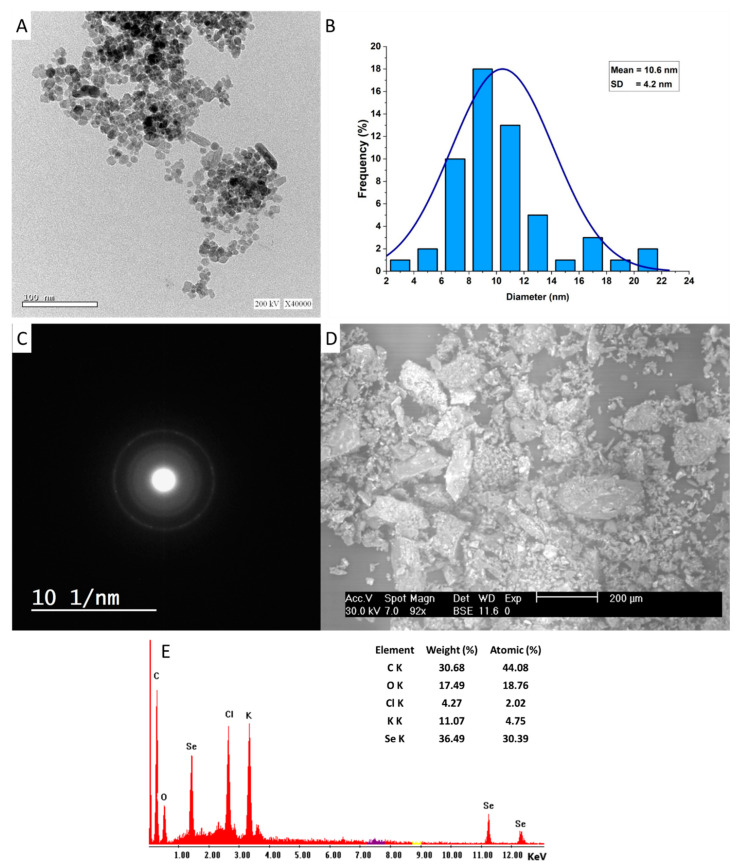
(**A**) TEM image of Se-NPs showing the spherical shape, (**B**) size distribution, (**C**) TEM-SAED, and (**D**,**E**) SEM-EDX showed the elemental compositions of the Phyto-synthesized Se-NPs.

**Figure 4 jfb-13-00157-f004:**
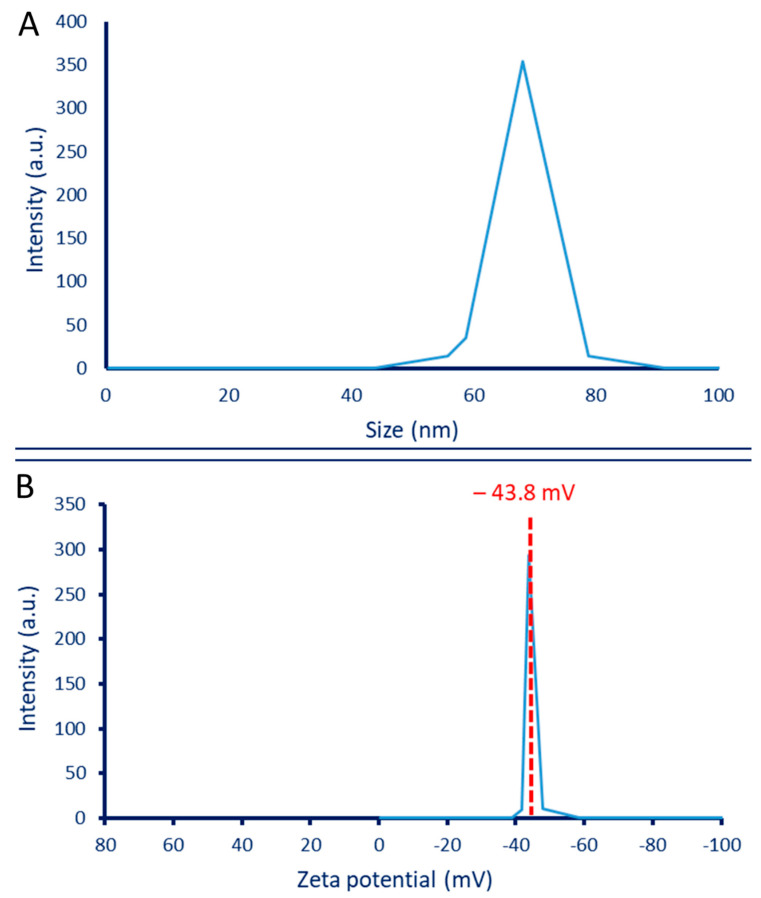
(**A**) Particle size analysis by dynamic light scattering (DLS), (**B**) zeta potential of the phyto-synthesized Se-NPs.

**Figure 5 jfb-13-00157-f005:**
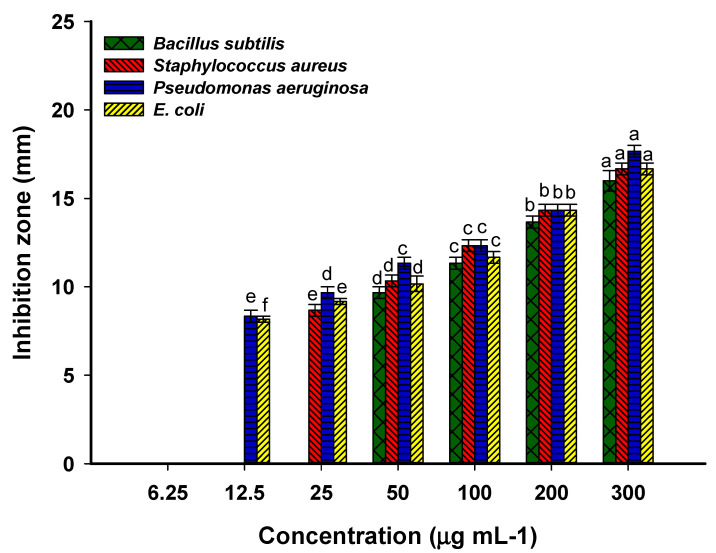
Antibacterial activity of the Phyto-synthesized Se-NPs against Gram-positive bacteria (*Bacillus subtilis* and *Staphylococcus aureus*) and Gram-negative bacteria (*Pseudomonas aeruginosa* and *Escherichia coli*). Different letters on the same column at different concentrations indicate the mean values between these concentrations are significantly different at *p* ≤ 0.05, *n* = 3.

**Figure 6 jfb-13-00157-f006:**
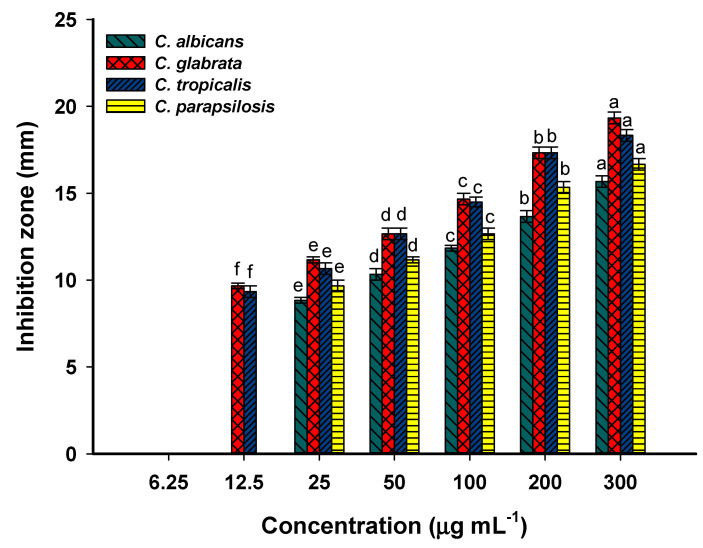
The efficacy of the Phyto-synthesized Se-NPs at different concentrations (300, 200, 100, 50, 25, 12.5, 6.25 µg·mL^−1^) against various clinical *Candida* strains (*C. albicans*, *C. glabrata*, *C. tropicalis*, and *C. parapsilosis*). Different letters on the same column at different concentrations indicate that mean values between these concentrations are significantly different at *p* ≤ 0.05, *n* = 3.

**Figure 7 jfb-13-00157-f007:**
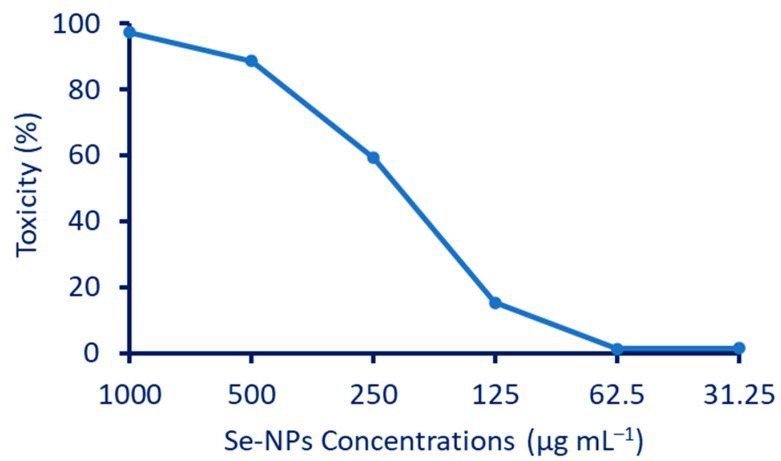
Toxicity percentages of Vero normal cells after treatment with various concentrations of Se-NPs (1000, 500, 250, 125, 62.5, and 31.25 µg·mL^−1^). The data are represented as mean ± SE (*n* = 3).

**Figure 8 jfb-13-00157-f008:**
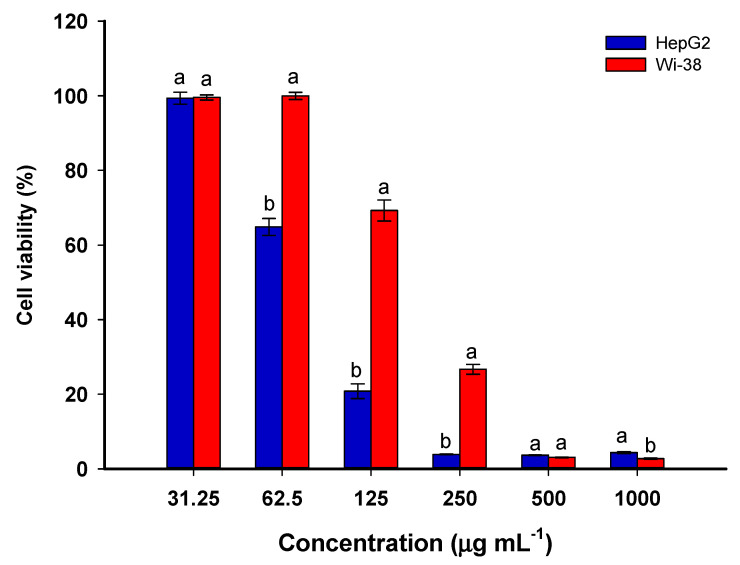
Cell viability percentages of WI-38 and HepG2 cells after treatment with various concentrations of Se-NPs. The data are represented as mean ± SE (*n* = 3). Different letters on the column at the same concentration indicate that mean values are significantly different (*p* ≤ 0.05).

**Figure 9 jfb-13-00157-f009:**
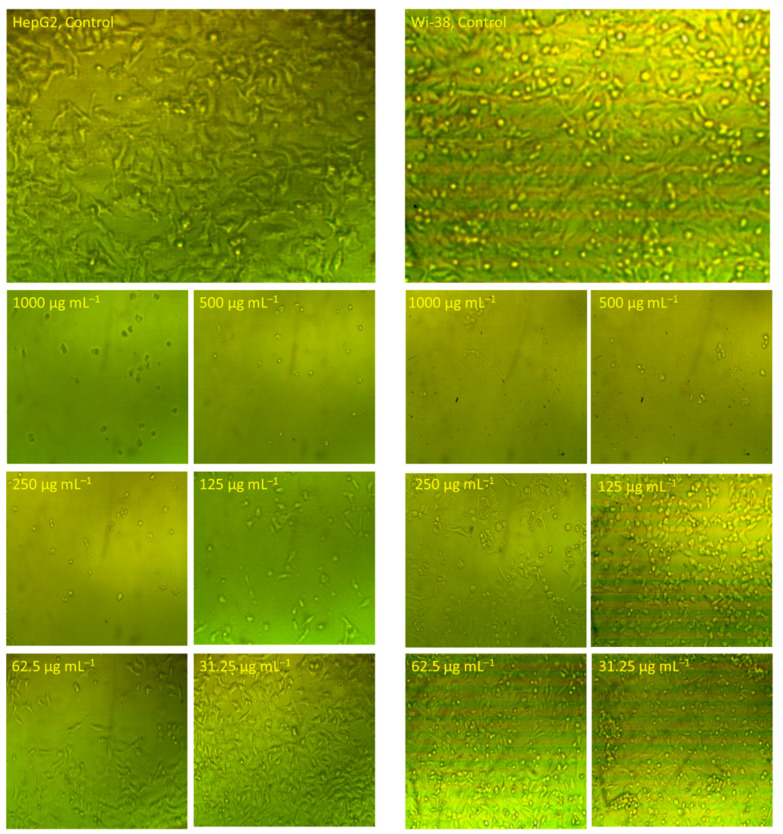
Morphological alterations of WI-38 and HepG2 cells in response to treatment with various concentrations of Se-NPs for 24 h under inverted microscopy. Magnification: 60×.

**Table 1 jfb-13-00157-t001:** Results of antiviral assay of non-lethal dose of Se-NPs (MNTC = 62.5 µg·mL^−1^), showing the percentages of viral activities and antiviral effects against HAV and Cox-B4 viruses.

Cell	Se-NPs (MNTC = 62.5 µg·mL^−1^)	Optical Density (OD) at 560 nm (Mean ± SE)	Viability (%)	Toxicity (%)	Viral Activity (%)	Antiviral Effect (%)
Vero cells (non-infected)	-	0.655 ± 0.010	100	0	-	-
HAV-infected (Vero cells)	-	0.309 ± 0.02	47.12	52.87	100	0
HAV-infected Vero cells + Se-NPs	62.5 µg·mL^−1^	0.552 ± 0.02	84.27	15.72	29.74	70.26
Cox-B4-infected with Vero cells	-	0.244 ± 0.01	37.30	62.70	100	0
Cox-B4-infected Vero cells + Se-NPs	62.5 µg·mL^−1^	0.501 ± 0.012	76.54	23.46	37.42	62.58

**Table 2 jfb-13-00157-t002:** Larvicidal activity of *P. oleracea*-based Se-NPs at different concentrations against various instar larvae (I, II, III, and IV) of *Culex pipiens*.

Larvae Stages	Concentrations (mg L^−1^)	Percentage Mortality	LC_50_ (LCL–UCL)	LC_90_ (LCL–UCL)	χ^2^ (*df* = 3)
I	Control	00.0 ± 0.0	1.12 (0.113–10.216)	42.4 (21.584–97.682)	6.521
	10	55.0 ± 2.0			
	20	60.2 ± 0.5			
	30	71.0 ± 1.0			
	40	77.7 ± 0.0			
	50	89.0 ± 3.1			
II	Control	00.0 ± 0.0	18.4 (7.274–49.237)	59.3 (24.689–101.287)	7.379
	10	35.0 ± 1.1			
	20	44.3 ± 1.0			
	30	50.0 ± 1.2			
	40	53.3 ± 1.0			
	50	73.0 ± 1.2			
III	Control	00.0 ± 0.0	25.4 (10.342–47.184)	66.4 (32.829–107.527)	8.468
	10	31.3 ± 1.5			
	20	37.3 ± 2.3			
	30	42.0 ± 1.0			
	40	52.0 ± 0.0			
	50	68.0 ± 1.4			
IV	Control	00.0 ± 0.0	28.8 (12.073–58.826)	69.4 (34.837–123.647)	9.521
	10	27.3 ± 1.5			
	20	30.3 ± 0.0			
	30	38.0 ± 1.0			
	40	42.0 ± 1.6			
	50	59.0 ± 1.0			

LC_50_ and LC_90_, lethal Se-NPs concentrations for 50% and 90% of larvae population; LCL and UCL, lower and upper confidence limit, respectively; χ^2^, chi-square value. Data are represented by mean ±SE *(n* = 5).

## Data Availability

The data presented in this study are available upon request from the corresponding author.

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
