# Peer review of "Antimicrobial, Antiviral, and In-Vitro Cytotoxicity and Mosquitocidal Activities of Portulaca oleracea-Based Green Synthesis of Selenium Nanoparticles"

_jfb, 2022, doi:10.3390/jfb13030157_

Round 1

Reviewer 1 Report

Please correct the following phrases to reduce the coincidence:

Line 52, coincidence with https://www.science.gov/topicpages/c/candida+drug+resistance

Lines 60-62, coincidence with https://www.dovepress.com/front_end/biomedical-potential-of-plant-based-selenium-nanoparticles-a-comprehen-peer-reviewed-fulltext-article-IJN

Line 92-96, 103-112, and 263-267, coincidence with https://www.nature.com/articles/s41598-022-15903-2

In general, I consider that the article presents exciting results that demonstrate the multitarget application of Se-NPs. In addition, the antecedents shown in the literature show the continuity of the line of work and the potential application of said particles. Finally, the characterization results are complete, and the FTIR spectra demonstrate their formation and the relevant functional groups if a deeper review of the grammar and writing is suggested regarding the citations of sources in the literature.

Author Response

Dear reviewer, 

Thank you very much for your valuable comments. We answered all comments point-by-point as shown in the author's response. 

Reviewer 2 Report

Major comments:

1-The constituents of the used plant extract  and their concentrations should be provided

2-XRD pattern is very poor and does not match the XRD of Se Nps I recommend collecting XRD again 

3-TEM-Selected area electron diffraction (TEM-SAED) of Se Nps should be added

4-SEM image of Se Nps has poor resolution and shows high agglomeration 

so, better high resolution SEM image should be provided

5-particle size analysis and distribution by size analysis instrument should be performed. also zeta potential should be specified

6-TGA analysis of Se Nps  should be performed

Author Response

(The authors gave the same response as above.)

Reviewer 3 Report

Dear authors

1. Pls state the level of significance below figures (you signed letters a, b etc but with no value stated for each letter).

2. How did you choosed cell lines? Why HepG2? And not some other? Since it seems that there is no logical explanation. Moreover, you have purchased bactereial strains but not Candida. In this case I strongly suggest to omit results regarding its antifungal properties, especially since you did not covered different fungal species!

3. Regarding Figure 9 - you did not address neighter the one of the proposed mechanisms and thus this figure may be omited from text.

4. This paper needs style correction!

3.

Author Response

(The authors gave the same response as above.)

Round 2

Reviewer 2 Report

The authors have addressed the previous comments and they have made sufficient changes. 

Author Response

Dear reviewer.

Thank you very much for your approval and your agreement.

Reviewer 3 Report

.

Author Response

Thank you for reviewing our manuscript and giving us helpful comments. We revised the paper according to your specific comments. Detailed explanations for the comments are shown below.

Reviewer comment: there are still some issues to be addressed in the present paper (e.g. authors have letters as significance in Figures. Several letters, but do not have different level of significance nor these letters address difference between different groups...). But this is minor point.

Author response: Thank you very much for your comment. The letters on the columns in each figure were clarified. For example, in Figure 5: this statement was added in the figure legend “Different letters on the same column at different concentrations indicate the mean values between these concentrations are significantly different at p≤0.05, n =3.” This means, for example, the letters “a, b, c, d, e, and f” on a column of E.coli refer to the clear zone formed due to treatment with “300, 200, 100, 50, 25, and 12.5 µg mL–1” are significantly different. Whereas the clear zone formed against Pseudomonas aeruginosa at a concentration of 100 and 50 µg mL–1 is not significant because it takes the same letter “c”.   This clarification will be the same for the other figures. We clarify this meaning in the legend of each figure.

Reviewer comment: My major concern is the amount f not connected data. All bacterial strains are from ATCC, but Candida is clinical isolate. I acknowledge the effort to screen potential effect of the substance but this manuscript has too many, non-related data! There is also antiviral screening, which I can interconnect to antibacterial and antifungal; but anticancer is way beyond all especially since authors choose two cell lines for which they claim to be good cancer and phisiological model!

I strongly suggest to divide the paper into two logical separate papers: e.g. one with description of particles + antiBACTERIAL effect and other to deal properly on its anticancer potential.

Author response: Thank you very much for your comment. The main hypothesis of the current study was to explore the multifunctional properties of green synthesized Se-NPs. Among these multifunctional properties are antibacterial, antifungal, antiviral, anticancer, and mosquitocidal. As shown, we are a group from different specializations, and each part in the manuscript serves a specific specialization. Moreover, the presence of these data (antibacterial, antifungal, antiviral, anticancer, and mosquitocidal activity with a physicochemical characterization of nanoparticles) can increase the reading and citation of the manuscript.

Finally, we hope the author response meets the approval.

Round 3

Reviewer 3 Report

.